# Single-nucleus RNA-sequencing in pre-cellularization *Drosophila melanogaster* embryos

**Ashley R. Albright**[1] *, **Michael R. Stadler**[2], **Michael B. Eisen**[2,3]

**1** Department of Biochemistry and Biophysics, University of California, San Francisco, San Francisco, California, United States of America, **2** Department of Molecular and Cell Biology, University of California, Berkeley, Berkeley, California, United States of America, **3** Howard Hughes Medical Institute, University of California, Berkeley, Berkeley, California, United States of America

* ashleyalbright93@gmail.com

**Data Availability Statement:** Raw sequencing files are available on Data Dryad: https://doi.org/10.6078/D13D9R All code is available here: https://github.com/aralbright/2021_AAMSME.

## Abstract

Our current understanding of the regulation of gene expression in the early *Drosophila melanogaster* embryo comes from observations of a few genes at a time, as with *in situ* hybridizations, or observation of gene expression levels without regards to patterning, as with RNA-sequencing. Single-nucleus RNA-sequencing however, has the potential to provide new insights into the regulation of gene expression for many genes at once while simultaneously retaining information regarding the position of each nucleus prior to dissociation based on patterned gene expression. In order to establish the use of single-nucleus RNA sequencing in *Drosophila* embryos prior to cellularization, here we look at gene expression in control and insulator protein, dCTCF, maternal null embryos during zygotic genome activation at nuclear cycle 14. We find that early embryonic nuclei can be grouped into distinct clusters according to gene expression. From both virtual and published *in situ* hybridizations, we also find that these clusters correspond to spatial regions of the embryo. Lastly, we provide a resource of candidate differentially expressed genes that might show local changes in gene expression between control and maternal dCTCF null nuclei with no detectable differential expression in bulk. These results highlight the potential for single-nucleus RNA-sequencing to reveal new insights into the regulation of gene expression in the early *Drosophila melanogaster* embryo.

## Introduction

Early animal development is largely driven by maternally-deposited RNAs and proteins. In *Drosophila melanogaster*, zygotic gene expression is detected as early as the 10th nuclear cycle; however, zygotic genome activation primarily occurs during the 14th nuclear cycle while maternal RNAs are degraded and cellularization begins [1, 2]. Much of the difficulty in understanding the regulation of early embryonic gene expression lies in the challenge to simultaneously capture expression level and patterning. Classic examples of patterned gene expression and regulation originate from *in situ* hybridizations [3–6], however the nature of *in*

**Funding:** ARA was supported by an NIH Training Grant (T21 GM 007127) and the National Science Foundation Graduate Research Fellowship Program. MRS was supported by an American Cancer Society postdoctoral fellowship (126730-PF-14-256-01-DDC). The work was also supported by a Howard Hughes Medical Institute Investigator award to MBE. The funders had no role in study design, data collection and analysis, decision to publish, or preparation of the manuscript.

**Competing interests:** MBE is a founder and former member of the board of directors of PLOS. This does not alter our adherence to PLOS ONE policies on sharing data and materials.

*situ* hybridizations does not allow for the study of many genes at once. In order to fully understand the regulation of gene expression across the genome, it is imperative that we establish new methods to examine changes in spatially-patterned genes.

Prior work from our lab demonstrated the use of RNA sequencing in patterning mutants following cryosectioning embryos across the anterior-posterior axis [7]. This work benefits from knowledge of the origin of each slice during analysis; however, with this method we cannot truly resolve from where the RNAs originated as many nuclei will contribute to expression within each slice. Recent work from Karaiskos, Whale, et al (2017) demonstrated the use of single-cell RNA-sequencing in the early *Drosophila* embryo and the ability to construct virtual *in situ* hybridizations from prior knowledge of patterned gene expression [8]. Others have used single-cell RNA-sequencing in dorsoventral mutant embryos and showed depletion of an entire subset of cells [9]. These studies demonstrate the potential for single-cell RNA-sequencing to answer questions relating to pattern and body axis formation in the early *Drosophila* embryo; however, whether single-cell RNA-sequencing is sensitive enough to detect subtle changes in gene expression in mutant embryos lacking major defects remains unclear.

To establish the use of single-nucleus RNA-sequencing in the early *Drosophila* embryo, we decided to examine gene expression in control, as well as maternal null dCTCF embryonic nuclei which are subsequently referred to as $dCTCF^{mat-/-}$. Insulator elements were first described for their enhancer-blocking activity [10–12], and have since been shown to affect genome and chromosome structure as well [13–18]. Interestingly, mammalian CTCF serves as the only insulator protein in mammals; however, *Drosophila* and other arthropods have evolved several insulator proteins [19, 20]. The redundancy of *Drosophila* insulator proteins allows us to understand the many functions of insulators without causing cell lethality. Intriguingly however, dCTCF is not actually required for embryonic viability [21]. Previous reports indicate that loss of individual *Drosophila* insulator proteins yields minimal changes in gene expression [19, 22–25], but others show that dCTCF is required for correct expression of certain genes observed by *in situ* hybridizations in embryos and larvae [26, 27]. The observed changes are slight however, which may explain why large-scale defects in transcription are not observed with RNA-sequencing in flies lacking dCTCF.

Using 10x Genomics, we assayed gene expression across over 8,000 nuclei from control and $dCTCF^{mat-/-}$ embryos. Overall, the nuclei tend to cluster according to expression of spatially-patterned genes, indicating that the nuclei retain information regarding their position in the embryo prior to dissociation. This allows us to understand genome-wide expression in spatial regions of embryos prior to cellularization by sequencing, which was previously only possible by slicing embryos [7, 13]. As expected considering the viability of $dCTCF^{mat-/-}$ embryos, we found fewer differentially expressed genes in bulk than in individual clusters. We also found several candidate patterned genes that may be differentially expressed in certain clusters but not in bulk. Our analyses are available in a reproducible and usable format (see Code Availability) allowing others to explore our data analysis as well as analyze other genes of interest not explored here. Altogether this work establishes the use of single-nucleus RNA-sequencing in the early *Drosophila* embryo to detect subtle changes in gene expression and encompasses a resource to explore candidate locally differentially expressed genes upon loss of maternal dCTCF.

## Methods

### CRISPR

Maternal dCTCF nulls were created by using CRISPR mutagenesis to insert a dsRed protein followed by two consecutive stop codons immediately upstream of the dCTCF open reading

frame. The homologous replacement template plasmid was constructed using a pUC19 backbone and ~1 kb homology arms generated by PCR (5' homology arm primers: CCACAAA-GAAACGTTAGCTAGTTCC and TCCTATGGACAAATTGGATTTGTTTTGG, 3' homology arm primers: CCAAGGAGGACAAAAAAGGACGAG and CGTGAGTGGCGCGTGATC). Repair template was coinjected into Cas9-expressing embryos (Rainbow Transgenic Flies, Camarillo, California), along with two guide RNAs (ATTTGTCCATAGGAATGCCA, TGTCCATAGGAATGCCAAGG) expressed from a U6:3 promoter on a modified version of the pCFD3 plasmid [28]). Resulting flies were crossed to flies containing chromosome 3 balancer chromosomes, and screened by genotyping PCR. Putative hits were further screened by PCR and sequencing of the entire locus using primers outside the homology arms (CATTAGAATT-CAAGGGCCATCAG and CACTTGAAGGATGGCTCG). A successful insertion line was recombined with an FRT site on chromosome 3L at cytosite 80B1 (Bloomington stock # BL1997).

## Fly husbandry

All stocks were fed standard Bloomington food from LabExpress and maintained at room temperature unless otherwise noted. We used the FLP-DFS (dominant female sterile) technique [29] to generate $dCTCF^{mat-/-}$ embryos. First, we crossed virgin *hsFLP, w\*;; Gl\*/TM3* females to *w\*;; ovo^D, FRT2A(mw)/TM3* males (Bloomington Drosophila Stock Center ID: 2139). From this cross, we selected *hsFLP,w\*;; ovo^D, FRT2A(mw)/TM3* males and crossed them to virgin *CTCF\*,FRT2A/TM3* females. Larvae from this cross were heat-shocked on days 4, 5, and 6 for at least two hours in a water bath at 37˚C. Upon hatching, virgin *hsFLP, w\*/+; CTCF\*, FRT2A (mw)/ovoD\*, FRT2A(mw)* females were placed into a small cage with their male siblings. Flies were fed every day with yeast paste (dry yeast pellets and water) spread onto apple juice agar plates. These crosses were conducted simultaneously with another insulator protein, and control embryos were collected from the *ovo^D* line used to generate those germline clones (Bloomington Drosophila stock Center ID: 2149).

## Western blots

Flies laid on grape-agar plates for two hours and embryos were either aged two hours at room temperature or taken directly after collection. Embryos were dechorionated with bleach, rinsed, and frozen in aliquots of ~25 embryos at -80 C. Embryos were homogenized in 25 μl RIPA buffer (Sigma cat # R0278) supplemented with 1 mM DTT and protease inhibitors (Sigma cat # 4693116001) using a plastic pestle. After homogenization, samples were mixed with 25 μl 2x Laemmli buffer (Bio-Rad # 1610737EDU), boiled for 3 minutes, and spun at 21,000 x g for 1 minute. Samples were loaded onto Bio-Rad mini Protean TGX 4–20% gels (# 4561096) and run at 200V for 30 minutes. Protein was transferred at 350 mA for one hour to Immobilon PVDF membrane (Millipore-Sigma # IPVH00010). Blots were blocked for one hour in PBST (1x PBS with 0.1% Tween) with 5% nonfat milk, and stained with primary antibodies (courtesy of Maria Cristina Gambetta [27], 1:1000 in PBST with 3% BSA) for one hour. Blots were then washed 3 times for 3 minutes rotating in PBST and probed with an HRP-conjugated anti-Rabbit secondary antibody (Rockland Trueblot, # 18-8816-33, 1:1000 in PBST with 5% milk) for one hour. After extensive washing with PBST, blots were developed with Clarity ECL reagents (Bio-Rad # 1705060) and imaged. Validation of the loss of maternal dCTCF is shown in S1 Fig.

## Nuclear isolation and sequencing

Nuclei were isolated from early to mid-nuclear cycle 14 embryos (stage 5) according to several previously published works [30–32]. First, the cages were cleared for 30 minutes to 1 hour to

remove embryos retained by the mothers overnight, followed by a 2 hour collection and 2 hour aging. Then, the embryos were dechorionated in 100% bleach for 1 minute, or until most of the embryos were floating, with regular agitation by a paintbrush. The embryos were transferred to a collection basket made of a 50 mL conical and mesh. After the embryos were rinsed with water, the embryos were transferred into an eppendorf tube containing 0.5% PBS-Tween. From this point forward, samples were kept on ice to prevent further aging of embryos.

A minimum of 9 early to mid-nuclear cycle 14 embryos were sorted using an inverted compound light microscope and transferred to a 2 mL dounce containing 600 uL of lysis buffer (10 mM 10 mM Tris-HCl pH 7.4, 10 mM NaCl, 3 mM MgCl2, 1% Bovine Serum Albumin, 1% RNase Inhibitor (Enzymatics, Part Num. Y9240L)) + 0.1% IGEPAL. The embryos were homogenized 20 times with a loose pestle and 10 times with a tight pestle. Pestles were rinsed with 100 uL lysis buffer + 0.1% IGEPAL after use. The resulting 800 uL of buffer and nuclei were transferred into an eppendorf tube, filtered with a 40 uM filter. Nuclei were pelleted by spinning for 5 minutes at 900 g and 4°C, washed in 500 uL lysis buffer (without 0.1% IGEPAL), and pelleted again before resuspending the nuclei in 20 uL lysis buffer (without 0.1% IGEPAL). Nuclei concentration was then adjusted to 1000 uL nuclei per uL, then nuclei were barcoded with the 10X Chromium Single Cell 3' Gene Expression Kit (v3). Control and $dCTCF^{mat-/-}$ nuclei were processed on separate days, then sequenced together with the Illumina NovaSeq (SP flow cell).

## Data processing and analysis

We used kallisto-bustools [33] to generate a custom reference index and generate a nucleus x gene matrix. The data were analyzed in both Python and R, using primarily scVI via scvi-tools [34, 35], scanpy [36], and custom scripts for analysis.

Control and $dCTCF^{mat-/-}$ nuclei were filtered separately as follows: (1) nuclei were ranked by the number of UMIs detected and nuclei ranked below the expected number of nuclei (10,000) were removed; (2) nuclei with fewer than 200 expressed genes were removed; (3) nuclei with greater than 5% mitochondrial expression were removed; (4) nuclei with greater than 50,000 UMI counts were removed; (5) genes expressed in fewer than 3 nuclei were removed.

Prior to batch correction, the data were subset to the 6000 most highly variable genes using scanpy's $dCTCF^{mat-/-}$ based on log1p normalized expression. We ran scVI with gene_likelihood = 'nb' to correct for batch effects.

The nuclei were clustered using the Leiden algorithm [37] within scanpy and visualized on a 2D UMAP [38]. Prior to batch correction, nuclei were clustered on log1p normalized gene expression. After batch correction, nuclei were clustered on the latent space derived from the scVI model. Marker genes representing each cluster were found using the sc.tl.rank_genes_groups function from scanpy with the Wilcoxon signed-rank test. *In situ* hybridizations of representative marker genes were obtained from the Berkeley Drosophila Genome Project[39–41]. Colors representing Leiden clusters were projected onto a virtual embryo using novoSpaRc [42, 43].

Log2 fold change and associated p-values were obtained for each gene using diffxpy (https://diffxpy.readthedocs.io/). Statistically significant differential expression was determined following Bonferroni correction of the p-values and filtered for adjusted p-value less than 0.05 and absolute value of log2 fold change greater than or equal to 1.5. Intersecting sets of differentially expressed genes were found and visualized with an UpSet plot [44, 45], following correction of adjusted p-values for the number of comparisons (multiplied by 11; 10 for the total number of clusters + 1 to include bulk differential expression).

## Data and code availability

Raw sequencing data and.h5ad files are available on DataDryad: https://doi.org/10.6078/D13D9R. Much of our analysis originated from work by Booeshaghi and Pachter (2020) [46] and Chari et al (2021) [47], with the addition of custom scripts.

All of the code used in the analysis and in generating the figures is available here: https://github.com/aralbright/2021_AAMSME. Single-nucleus data pre-processing, batch correction and clustering, virtual *in situ* hybridization, and differential expression analyses are available in this GitHub repository as Google Colab notebooks. These notebooks are available for anyone to run from a web browser with the option to enter any genes of interest not discussed in this manuscript.

## Results

To establish the use of single-nucleus RNA-sequencing for examining gene expression in the early *Drosophila* embryo prior to cellularization, we hand-sorted 10 to 20 early to mid-nuclear cycle 14 control and $dCTCF^{mat-/-}$ embryos and isolated nuclei for single-nucleus RNA-sequencing using 10x Genomics 3' Gene Expression. After filtering the data for high quality nuclei and correcting for non-biological variability (S2–S4 Figs), we used Leiden clustering [37] to detect distinct groups of nuclei from control and $dCTCF^{mat-/-}$ embryos, altogether resulting in 8,400 nuclei across 10 clusters (Fig 1A) composed of both control and $dCTCF^{mat-/-}$

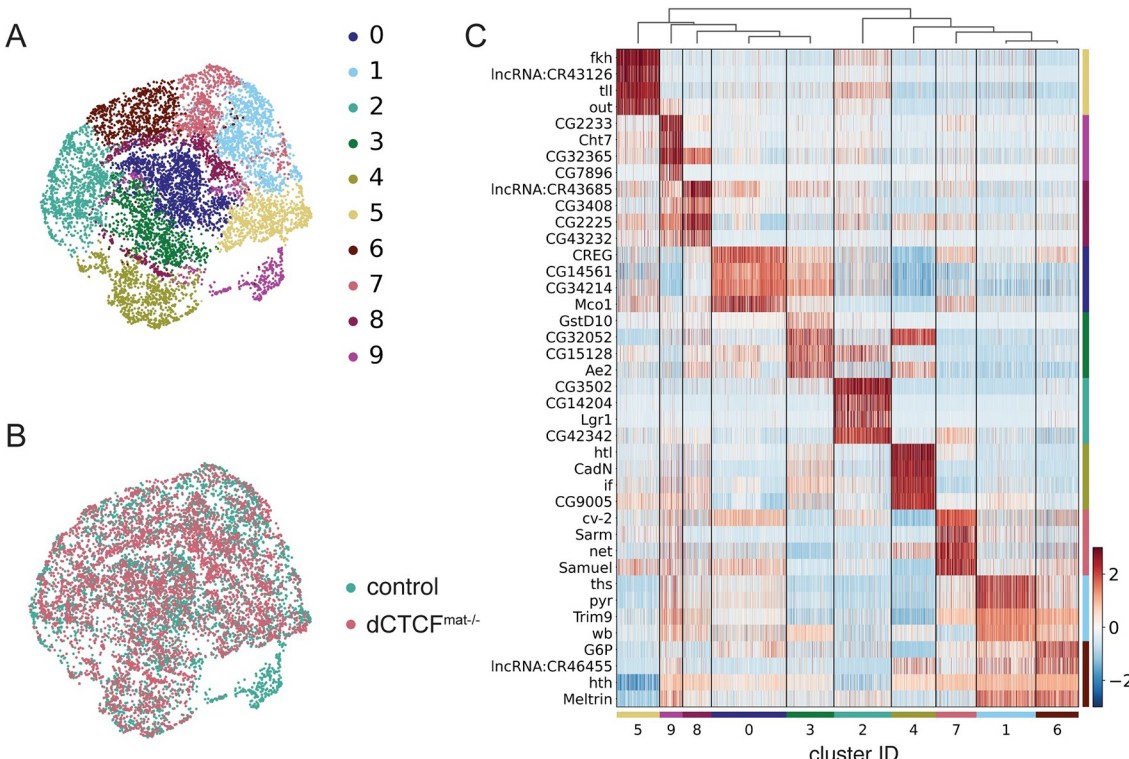

**Fig 1.** Single-nucleus RNA-sequencing analysis of pre-cellularization *Drosophila melanogaster* embryos (A) Two-dimensional UMAP embedding of nuclei shows 10 transcriptionally distinct clusters, as determined using the Leiden algorithm. Associated colors are maintained throughout the manuscript. (B) Two-dimensional UMAP embedding of nuclei labeled by condition shows the overlap of control and $dCTCF^{mat-/-}$ nuclei in a reduced dimensional space following correction for non-biological variability. Associated colors are maintained throughout the manuscript. (C) Heatmap of scaled gene expression for top four marker genes of each cluster, clusters are hierarchically ordered.

nuclei (Fig 1B). We also removed yolk and pole cell nuclei as these nuclei are not informative for patterned gene expression; however, the fact that subsets of nuclei clustered on marker gene expression for yolk or pole cell nuclei provided us with confidence that our data accurately represent single-nucleus expression (S5 Fig). After removing groups of nuclei as indicated, the nuclei no longer cluster according to expression of yolk or pole cell markers, indicating that our data are of high quality (S6 Fig). Once we finalized the dataset, we then asked whether gene expression in the clusters determined by the Leiden algorithm are truly distinct.

Given how well characterized patterned gene expression is in the early *Drosophila* embryo and that we found several distinct clusters of nuclei, we suspected that the clusters may represent different spatial regions within the embryo. Expression of the top marker genes representing each cluster is certainly distinct, and we noticed that many of these genes are expressed in patterns, namely *fkh*, *tll*, and *htl* (Fig 1C). To determine if single-nucleus RNA-sequencing in the early embryo can be spatially resolved, we examined *in situ* hybridizations of top marker genes for each cluster. We found that representative virtual and published *in situ* hybridizations of the top 20 marker genes (S1 Table) correspond to specific spatial regions within the embryo for clusters 0–7 (Fig 2A–2H and 2A'–2H'). We also found by projecting our nuclei onto a virtual embryo that the identities we assigned to each cluster correspond to the spatial identities of these clusters (Fig 2I).

The anterior, posterior, and ventral clusters are the most defined based on a projection of our nuclei onto a virtual embryo, while clusters that represent the middle of the embryo in general had less well defined borders (Fig 2I). Even so, our data shows that single-nucleus RNA-sequencing in the early *Drosophila* embryo yields information related to the spatial position of nuclei prior to dissociation. We should note however, that the virtual *in situ* hybridizations shown above, as well as additional virtual *in situ* hybridizations that represent each cluster, contain genes present in the list of reference genes used to generate the virtual patterns (see *Ilp4*, *htl*, *fkh* in Fig 2, and *Antp*, *NetA*, *disco* in S7 Fig). As such, we considered that the virtual *in situ* hybridizations may be biased in those cases; however, we believe the presence of several other genes representing each cluster with similar patterning validated with both virtual and published *in situ* hybridizations indicates that reference bias is not an issue. In the end, we were unable to determine a spatial identity for clusters 8 and 9; however, we decided to include these clusters in subsequent analyses because the nuclei passed our quality control filters. Interestingly, cluster 9 appears to be absent in *dCTCF*^mat-/-^ embryos (Fig 1A and 1B). Without knowing the identity of cluster 9, we can only speculate why this may be the case; however, this raises the possibility that dCTCF may play a role in nuclear fate.

In an effort to establish the use of single-nucleus RNA-sequencing to detect local changes in gene expression in embryos prior to cellularization, we then asked whether we could detect potential differential expression of genes in individual clusters, but not in bulk. In most clusters and in bulk, gene expression appears to be up-regulated upon loss of maternal dCTCF (S8 Fig). We also found that differentially expressed genes shared between all clusters and in bulk represent one of the largest shared sets (Fig 3A, left most black bar). However, a substantial number of candidate differentially expressed genes appear differentially expressed in single clusters (Fig 3A, blue bars). Many other genes appear differentially expressed in groups of clusters, but not in bulk. Because we found many candidate differentially expressed genes, we considered that this may be due to low expression given the sparsity of single-nucleus RNA-sequencing; however, we found that the mean expression of candidate differentially expressed genes in single or multiple clusters overall does not have a substantially different pattern from that of non-differentially expressed genes (S9 Fig). Each of these curves are right-skewed, or most genes are expressed in low levels at less than 100 transcripts per million (TPM).

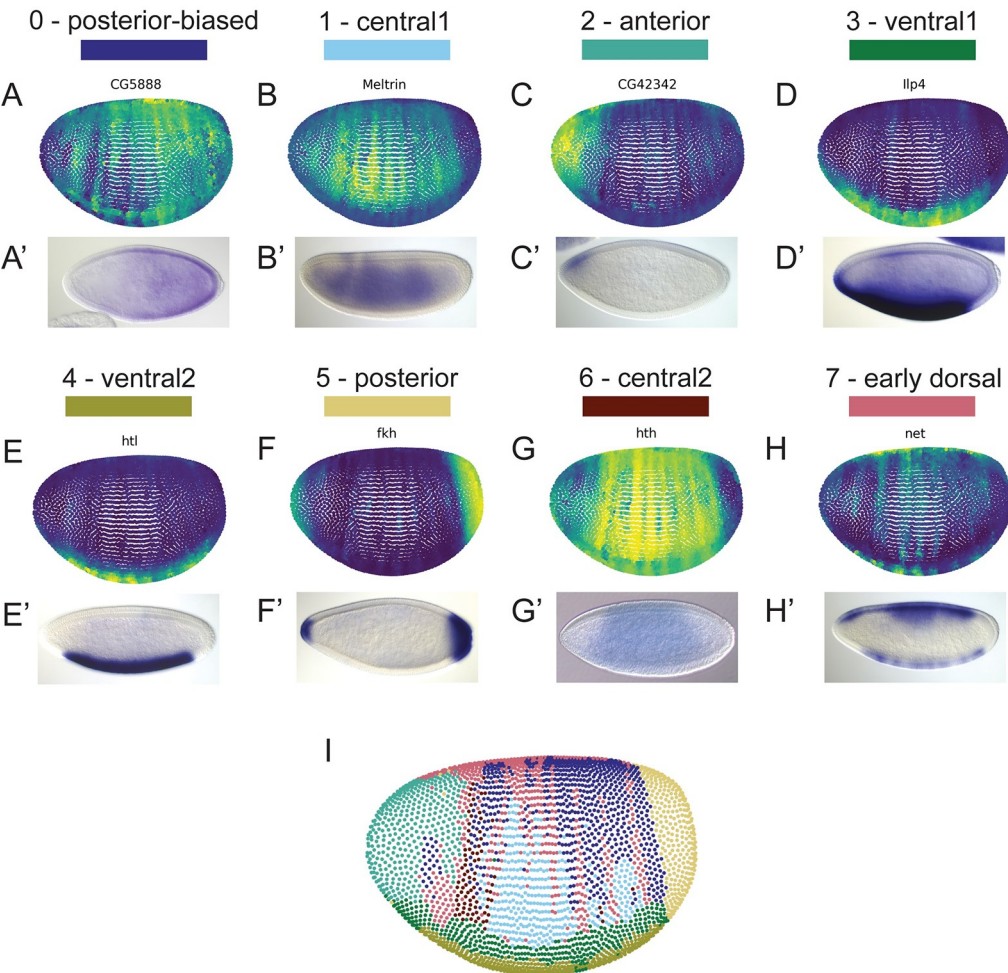

**Fig 2.** Leiden clusters correspond to spatial regions within the embryo (A-H) Representative virtual *in situ* hybridizations (top, A-H) for top marker genes representing each cluster as labeled and the corresponding published *in situ* hybridizations (bottom, A'-H') from the Berkeley Drosophila Genome Project [39–41]. (I) Projection of nuclei onto a virtual embryo labeled by the Leiden cluster as colored in Fig 1A. Virtual *in situ* hybridizations and projection of clusters onto a virtual embryo were generated using novoSpaRc [42, 43].

Altogether, these results show that single-nucleus RNA-sequencing in the early embryo can be used to detect candidate differentially expressed genes that would not appear in bulk-sequencing data.

Upon the loss of an early developmental factor like dCTCF, we expect to observe potential differential expression of patterned genes in specific clusters with single-nucleus RNA-sequencing. Interestingly, *stumps*, a ventrally-expressed gene is differentially expressed in one ventral cluster, but not the other (Fig 3B). *bowl*, a gap gene primarily expressed in the anterior, appears to be up-regulated in the posterior-biased and one of the ventral clusters (Fig 3C). Finally, *Esp*, a posterior-striped gene is differentially expressed in several clusters. Intriguingly, we did not detect differential expression in bulk for *stumps, bowl, or Esp*. Because dCTCF is required for proper expression of *Abd-B [26]*, a *Drosophila* Hox gene, we also examined expression of several *Drosophila* Hox genes within each cluster and in bulk. Upon the loss of maternal dCTCF, *Antp* and *abd-A* are potentially differentially expressed in certain clusters, with potential differential expression of *Antp* in bulk data (S10 Fig). However despite the

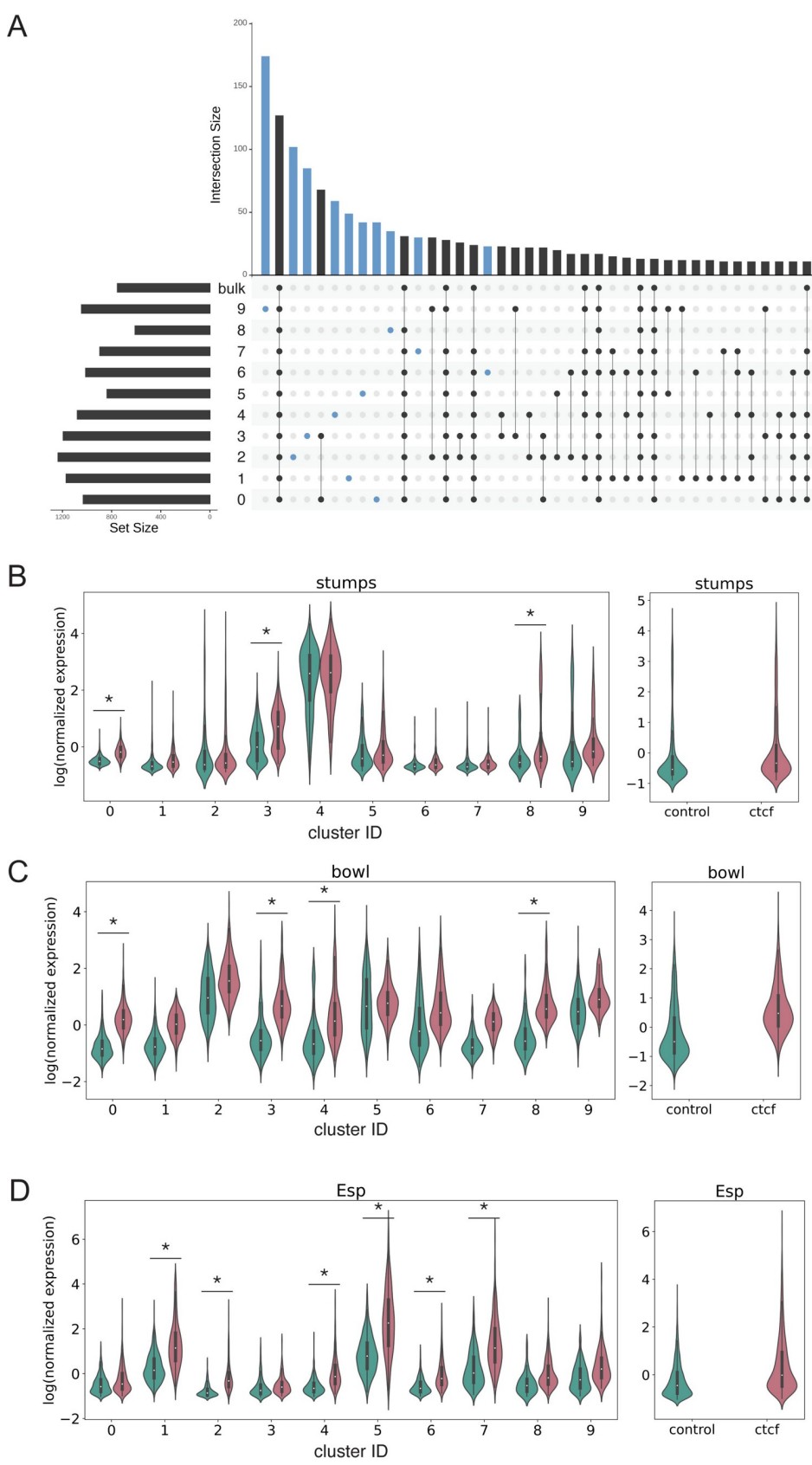

**Fig 3.** Differential expression of genes detected in one or more clusters, but not in bulk (A) UpSet plot for visualizing the top 40 shared sets of candidate differentially expressed genes between control and *dCTCF*<sup>*mat-/-*</sup> nuclei within each cluster and in bulk. Horizontal bar plot (A, left) represents the total number of candidate differentially expressed genes within the cluster of the corresponding row. The vertical bar plot (A, top) represents the number of shared candidate differentially expressed genes for the conditions indicated below and is sorted from largest to smallest intersecting set, with each count representing a unique gene. Connected dots (black) represent the corresponding group of genes in the vertical bar plot above that might be differentially expressed in the clusters represented by rows with a filled in circle. Candidate genes differentially expressed in a single cluster are represented in blue. (B-D) log(scvi normalized expression) of (B) *stumps*, (C) *bowl*, (D) *Esp* in each cluster (left) and bulk (right) for control (teal) and *dCTCF*<sup>*mat-/-*</sup> nuclei (pink). Asterisks indicate statistically significant differential expression (absolute value of expression $\geq 1.5$ and Bonferroni corrected p-value $< 0.05$).

requirement of dCTCF for proper expression of *Abd-B* shown by *in situ* hybridization [26], we found no evidence of differential expression of *Abd-B*, in agreement with bulk RNA sequencing in larval CNS dCTCF mutants [27].

We cannot be certain whether or not specific genes are truly differentially expressed spatially without further investigation; however, our results demonstrate the use of single-nucleus RNA-sequencing to detect possible local changes in gene expression upon perturbation in the early embryo prior to cellularization.

## Discussion

We conducted the above analyses in order to determine whether we could use single-nucleus RNA-sequencing as a means of understanding the regulation of gene expression in the early *Drosophila* embryo. First, we show that nuclei can be grouped into clusters represented by distinct gene expression. Then, we show that representative marker genes from the majority of the clusters recapitulate known patterns of expression. Importantly, we also present examples of potential differential expression of patterning genes in individual clusters upon loss of maternal dCTCF, but not in bulk.

Prior to this work, studies towards our understanding of the regulation of patterned gene expression in a spatial context included cytoplasmic RNAs in measures of expression. We must acknowledge the caveat that we do not know the extent to which maternal RNAs enter the nucleus and some of our results may reflect the presence of both maternal and zygotic RNAs. Nonetheless, we believe that single-nucleus RNA-sequencing is better suited as opposed to bulk RNA-sequencing to understand changes in gene expression in pre-cellularization embryos upon the loss of important developmental factors because of the ability to resolve local changes in expression. Supporting this notion, single-cell RNA-sequencing has already shown to resolve the loss of an entire cell fate in cellularized dorsoventral mutant embryos [9].

Whether or not the changes in gene expression that we observed have implications in embryonic development related to the loss of dCTCF is unclear without further investigation, such as single-molecule FISH to validate the observed changes in gene expression of particular RNAs. Ultimately, using single-nucleus RNA-sequencing to examine changes in gene expression upon the loss of important developmental factors has the potential to uncover perturbation responses previously undetected by bulk RNA-sequencing.

## Supporting information

**S1 Table. Top 20 marker genes representing each cluster as determined by sc.tl.rank_genes_groups.**
(CSV)

**S1 Fig.** (A) Western blotting of OreR, 0h and 2h *dCTCF*<sup>*mat-/-*</sup> embryos using an antibody to *Cp190*, another insulator protein, as a control. (B) Western blotting of OreR, 0h and 2h *dCTCF*<sup>*mat-/-*</sup> embryos using an antibody to *dCTCF*. The 2h embryos were aged for an additional 2 hours with the majority of the embryos representing nuclear cycle 14, the same time point at which we conducted single-nucleus RNA-sequencing. A cross-reactive band appears at approximately 75 kd, with the *dCTCF* band appearing at approximately 130 kd.
(TIF)

**S2 Fig.** (A) Knee plot for barcodes ranked by the number of UMIs versus UMI counts for control (left) and *dCTCF*<sup>*mat-/-*</sup> (right) experiments. Black line indicates the position of the 10,000<sup>th</sup> (expected number of cells) on each axis. (B) Percent mitochondrial expression per nucleus in control (left) and *dCTCF*<sup>*mat-/-*</sup>(right) nuclei. Dashed line represents 5% mitochondrial expression, or the cutoff used for filtering the data. (C) Number of genes detected per nucleus by UMI counts in control (left) and *dCTCF*<sup>*mat-/-*</sup>(right) nuclei.
(TIF)

**S3 Fig.** Number of genes detected (left), UMI counts (middle), percent mitochondrial expression (right) per nucleus after filtering in (A) control and (B) *dCTCF*<sup>*mat-/-*</sup> experiments.
(TIF)

**S4 Fig.** Two-dimensional UMAP embedding of control (teal) and dCTCFmat-/- (pink) nuclei (A) before, (B) after batch correction using scVI, and (C) after removing low quality nuclei.
(TIF)

**S5 Fig.** Two-dimensional UMAP embedding of nuclei before additional filtering colored by (A) number of genes detected, (B) UMI counts, (B) percent mitochondrial expression. (D-F) log(scvi normalized expression) of three genes with representative *in situ* hybridizations below for (D) cell cycle gene *aurB*, (E) yolk nucleus marker, *sisA* (F) and pole cell marker *pgc*.
(TIF)

**S6 Fig.** Two-dimensional UMAP embedding of nuclei after removal of clusters with high percent mitochondrial expression, *aurB* expression, *sisA* expression, and *pgc* expression colored by (A) number of genes detected, (B) UMI counts, (C) percent mitochondrial expression. (D-F) log(scvi normalized expression) of three genes with representative *in situ* hybridizations below for (D) cell cycle gene *aurB*, (E) yolk nucleus marker, *sisA* (F) and pole cell marker *pgc*.
(TIF)

**S7 Fig.** (A-P) Representative virtual (top, A-P) and Berkeley Drosophila Genome Project (bottom, A'-P') *in situ* hybridizations for additional marker gene expression within each cluster as indicated. This supplemental figure accompanies Fig 2 in the main text.
(TIF)

**S8 Fig.** Volcano plots of log2FC (log2(fold-change)) by the log of the adjusted p-value (p-adj) for differential expression calculated in bulk (top middle) and in individual clusters as indicated. Colored dots indicate genes with significant differential expression, an absolute value of log2FC $>= 1.5$ and p-adj $< 0.05$. Significantly down-regulated genes are indicated in green, significantly up-regulated genes in pink, and non-significantly differentially expressed genes in light gray.
(TIF)

**S9 Fig. Histogram of average gene expression in transcripts per million (TPM) of differentially expressed genes in one cluster (yellow), differentially expressed in multiple clusters and/or in bulk (blue), and non-differentially expressed genes (red).** Each count on the y-

axis represents a single gene.
(TIF)

**S10 Fig.** Plots of log(scvi normalized expression) of select (A) Antennapedia complex genes: *Dfd* (top), *Scr* (middle), and *Antp* (bottom) and (B) Bithorax complex genes: abd-A (top) and Abd-B (bottom) in each cluster (left) and in bulk (right) for control (teal) and $dCTCF^{mat-/-}$ nuclei (pink). Asterisks indicate statistically significant differential expression (absolute value of expression $> = 1.5$ and Bonferroni corrected p-value $< 0.05$).
(TIF)

## Acknowledgments

We are grateful for Maria Cristina Gambetta and the generous sharing of Cp190 and dCTCF antibodies. Thank you to Dr. Justin Choi and the UC Berkeley Functional Genomics Laboratory as well as the Vincent Coates Sequencing Laboratory for processing our samples. We would also like to thank Sina Booeshaghi, Tara Chari, and Lior Pachter of the Pachter Lab at Caltech for their feedback and discussions on single-nucleus RNA-sequencing analysis. We thank Colleen Hannon and Marc Singleton and all members of the Eisen lab for helpful discussions and advice throughout preparation of the manuscript.

## Author Contributions

**Conceptualization:** Ashley R. Albright.

**Data curation:** Ashley R. Albright.

**Formal analysis:** Ashley R. Albright.

**Funding acquisition:** Ashley R. Albright, Michael R. Stadler, Michael B. Eisen.

**Investigation:** Ashley R. Albright, Michael R. Stadler.

**Methodology:** Ashley R. Albright.

**Project administration:** Ashley R. Albright.

**Validation:** Ashley R. Albright.

**Visualization:** Ashley R. Albright.

**Writing – original draft:** Ashley R. Albright.

**Writing – review & editing:** Ashley R. Albright, Michael R. Stadler, Michael B. Eisen.

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
