## [Decision Letter · Decision Letter 0]

16 Feb 2022

PONE-D-22-01112Single-nucleus RNA-sequencing in pre-cellularization Drosophila melanogaster embryosPLOS ONE

Dear Dr. Albright,

Thank you for submitting your manuscript to PLOS ONE. Your study has now been evaluated by two reviewers. As you will see both reviewers find your work of interest but also raise several points that would need to be addressed before publication can be considered. The reviewers point out in their comments where clarifications are needed and additional analysis would be required for further supporting your conclusions. I therefore, return your manuscript to you for a revision. Please, note that the revised version will be sent to the reviewers and a decision for publication made on their assessment. 

We look forward to receiving your revised manuscript.

Kind regards,

Anton Wutz

Academic Editor

PLOS ONE

Journal Requirements:

"I have read the journal's policy and the authors of this manuscript have the following competing interests: MBE is a founder and former member of the board of directors of PLOS."

Reviewers' comments:

Reviewer's Responses to Questions

**Comments to the Author**

1. Is the manuscript technically sound, and do the data support the conclusions?

Reviewer #1: Yes

Reviewer #2: No

2. Has the statistical analysis been performed appropriately and rigorously? 

Reviewer #1: Yes

Reviewer #2: No

3. Have the authors made all data underlying the findings in their manuscript fully available?

Reviewer #1: Yes

Reviewer #2: No

4. Is the manuscript presented in an intelligible fashion and written in standard English?

Reviewer #1: Yes

Reviewer #2: Yes

5. Review Comments to the Author

Reviewer #1: This study presents results of single-nucleus RNA-sequencing (snRNA-seq) in early fly embryos undergoing zygotic genome activation. Wildtype (WT) and CTCF[mat-] mutants lacking maternal CTCF were both analyzed. Transcripts detected in single WT nuclei could be mapped onto a virtual reference embryo using known marker genes, and recapitulated known spatial expression patterns similarly to single-cell RNA-seq (Karaiskos et al. 2017). Differential gene expression between WT and CTCF[mat-] embryos was analyzed. Measuring transcript abundance differences between individual snRNA-seq clusters in WT and mutant embryos identified more differentially expressed genes than measuring transcript abundance differences between all cells of WT and mutant embryos in bulk.

These results are interesting because they show that snRNA-seq is sensitive enough to detect relatively subtle gene misexpression defects in mutant embryos lacking major defects in cell fate decisions. The data appears to be of high quality. But I have 2 major confusions about the study’s design that must be addressed prior to publication (see major comments).

Major comments:

1. Please clearly describe whether the CTCF[mat-] embryos generated in this study zygotically express CTCF, and discuss whether this confounds analyses of differential gene expression in CTCF[mat-] embryos if these embryos already initiated zygotic transcription.

2. I am confused by line 59. How can snRNA-seq help understand how gene expression is established prior to zygotic genome activation? If the zygotic genome is not transcribed, what would snRNA-seq detect?

3. Fig. S1 is missing.

Minor comments

4. Line 40 is not well written: “Much of the difficulty in understanding the regulation of early embryonic gene expression lies in our ability [in the challenge?] to simultaneously capture expression level and patterning”.

5. Line 74-75: Kaushal et al. 2021 Nat Comm report differentially expressed genes in CTCF mutants lacking maternal and zygotic CTCF relative to WT by RNA-sequencing. This seems contradictory with the statement that differential gene expression in CTCF mutants has “not been found via sequencing”.

6. Lines 80-81: “Differential expression in spatial regions by sequencing” was previously performed by the references cited in the introduction, and therefore stating that this “was previously only possible by mechanical manipulation” should be toned down.

7. Line 169 must be amended (“nuclei ranked below the to the expected number of nuclei”).

8. Line 206: Figure references have typos.

9. Lines 218 and 220: First words of sentences should be capitalized.

10. I did not understand the meaning of the sentence in lines 234-236.

11. Cluster 9 should be discussed further, even if its spatial identity could not be determined. Why is this cluster only detected in WT but not CTCF[mat-] embryos?

12. Fig. S4 panel C is not described in the figure legend.

Reviewer #2: The authors Albright et al classified embryonic nuclei by single-nucleus RNA-seq and examined CTCF-regulated gene expression in these nuclei by comparing wild type to CTCF maternal null mutants during zygotic genome activation. They identified more cluster-specific differential expression than in bulk RNA-seq and highlighted several examples of differential expression of spatial marker genes in specific clusters. The work should be of general interest. However, the data requires further analyses, and the conclusions were not clearly presented or fully supported. Major revisions are needed to both the analyses and writing.

1. Figure numbering is incorrect for all supplementary figures, and it is not possible to understand which figures the authors were calling for. The intended Figure S1 is missing. There is no Figure S9 in the submission.

2. The authors should indicate whether there is zygotic CTCF expression in this mutant. A diagram will be very helpful. The missing Figure S1 makes it more challenging to understand the properties of this KO.

3. Although the CTCF maternal knockout is known to be viable, dysregulation of Hox gene expression has been reported in embryos. The authors should characterize their new mutants and compare with previous data (Gambetta and Furlong, 2018) to report consistent or distinct phenotypes, e.g., viability and expression pattern of Hox genes.

4. Figure 1B indicates loss of cluster 9 in CTCF maternal KO. Is this because the cells are absent or because their gene expression changes and they are classified into other clusters? This can be determined by in situ of cluster 9-specific genes.

5. What is the accuracy of spatial prediction based on the RNA-seq? How many top marker genes were checked, and how many have consistent expression patterns with the in situ data?

6. The authors should verify their knockout by western blots, which are mentioned but not presented.

7. The authors stated that spatial identities cannot be assigned to clusters 8 and 9, but some quick searches with gene IDs in Figure 1C identified embryonic CNS for cluster 8 (maybe 9) and another clear distinct spatial pattern for cluster 9. The authors need to dig deeper into the spatial identity by searching more genes in those clusters using publicly available data.

http://www.flyexpress.net/search/genes/CG2233/images/BDGP/LDVO

http://www.flyexpress.net/search/genes/CG2225/images/BDGP/LDVO

http://www.flyexpress.net/search/genes/CG3408/images/BDGP/LDVO

8. The authors need to indicate how much overlap there is between differential expression of different clusters. For example, cluster 9 has ~170 DE, cluster 3 has ~80 DE, but the common DE between clusters 3 and 9 is ~20 genes. Does this mean that most DE genes are cluster-specific? The authors need to quantify and present the data clearly. It is extremely hard to decipher the information presented in current Figure 3A.

9. The authors concluded that they identified more cluster-specific DE than in bulk but did not present data beyond the information in Figure 3A, horizontal bars. Do these bars include primarily the same genes, or they are totally different genes? It is very unclear how many more genes are detected as cluster-specific than in bulk. The authors state on page 12, “Many other genes are also differentially expressed in groups of clusters, but not in bulk.” This is very important and quantifiable information, and the authors need to present the data clearly.

10. The authors need to verify their RNA-seq data with another type of assay, e.g., RNA FISH. The differential expression (e.g., Figure 3B and C) should be verified between wildtype vs. CTCF KO cells, and between wildtype clusters of cells. Without such verification, it is hard to conclude that the single nucleus RNA-seq provides useful information. The verification should be done at least for the top differences.

11. How do the authors explain that most DE is up-regulation? Does this agree with Kaushal et al., 2021? It seems likely that decreases in DE are still masked by the inability to further separate cell types using this approach.

12. Lines 269-272: “Because we found many differentially expressed genes, we considered that this may be due to low expression given the sparsity of single-nucleus RNA-sequencing; however, we found that the mean expression of differentially expressed genes in single or multiple clusters overall does not have a substantially different pattern from that of non-differentially expressed genes.” The authors should compare and present mean expression of differential genes versus non-differential genes. This analysis is essential to rule out the possibility that more differential genes in cluster data than bulk data result from inaccurate quantification due to insufficient sequencing and coverage. Figure S9 may contain such information but is currently missing. The authors should also clearly define what they mean by “a substantially different pattern”.

13. Figure 3B-D: the expression change of patterning markers may lead to morphogenesis defects – did the authors examine the tissue morphology and distribution of marker gene expression by in situ or RNA FISH in embryos/larvae? Is the difference caused by strong differences in a small group of cells or weak differences in all cells in one cluster?

14. What are the most affected factors and signaling pathways in CTCF KO? How many of these genes are CTCF binding targets based on published CTCF ChIP-seq?

15. Please finish the sentence in lines 141, 169 (“to” and “the expected number of *UMI*”?).

6. PLOS authors have the option to publish the peer review history of their article (what does this mean?). If published, this will include your full peer review and any attached files.

Reviewer #1: No

Reviewer #2: No

---

## [Author Response · Author response to Decision Letter 0]

17 May 2022

In the spirit of PloS we wanted to provide an open access resource from a rigorous experiment for others to use in future work. Regarding the availability of our data and analysis, we realized after receiving this review that some of the links in our Google Colab notebooks were broken. We fixed this issue prior to the submission of revisions. 

5. Review Comments to the Author

Reviewer #1: This study presents results of single-nucleus RNA-sequencing (snRNA-seq) in early fly embryos undergoing zygotic genome activation. Wildtype (WT) and CTCF[mat-] mutants lacking maternal CTCF were both analyzed. Transcripts detected in single WT nuclei could be mapped onto a virtual reference embryo using known marker genes, and recapitulated known spatial expression patterns similarly to single-cell RNA-seq (Karaiskos et al. 2017). Differential gene expression between WT and CTCF[mat-] embryos was analyzed. Measuring transcript abundance differences between individual snRNA-seq clusters in WT and mutant embryos identified more differentially expressed genes than measuring transcript abundance differences between all cells of WT and mutant embryos in bulk.

These results are interesting because they show that snRNA-seq is sensitive enough to detect relatively subtle gene misexpression defects in mutant embryos lacking major defects in cell fate decisions. The data appears to be of high quality. But I have 2 major confusions about the study’s design that must be addressed prior to publication (see major comments).

Major comments:

1. Please clearly describe whether the CTCF[mat-] embryos generated in this study zygotically express CTCF, and discuss whether this confounds analyses of differential gene expression in CTCF[mat-] embryos if these embryos already initiated zygotic transcription.

We have addressed this comment by adding the following statement at the end of the first paragraph in the ‘Results’ section: “We confirmed that our dCTCFmat-/- embryos lack maternal dCTCF at 0h and 2h after laying via western blot (S1 Fig). These experiments were conducted agnostic of the zygotic genotype given that zygotic genome activation largely does not occur until nuclear cycle 14 [1,2].”

2. I am confused by line 59. How can snRNA-seq help understand how gene expression is established prior to zygotic genome activation? If the zygotic genome is not transcribed, what would snRNA-seq detect?

To address this comment, we removed the following text: ‘zygotic genome activation.’ While it’s true that large-scale zygotic genome activation does not occur until nuclear cycle 14 (with a subset of genes expressed before then)… the true need for snRNA-seq lies in the fact that the embryo is not composed of cells at this point. 

3. Fig. S1 is missing.

We thank you for pointing out this mistake and have corrected the figure submissions. 

Minor comments

4. Line 40 is not well written: “Much of the difficulty in understanding the regulation of early embryonic gene expression lies in our ability [in the challenge?] to simultaneously capture expression level and patterning”.

We replaced “in our ability” with “in the challenge” as suggested. 

5. Line 74-75: Kaushal et al. 2021 Nat Comm report differentially expressed genes in CTCF mutants lacking maternal and zygotic CTCF relative to WT by RNA-sequencing. This seems contradictory with the statement that differential gene expression in CTCF mutants has “not been found via sequencing”.

Kaushal et al. 2021 found differential expression in 386 genes upon loss of CTCF, only 10% of which had a CTCF peak within 1 kb of the transcription start site. This enrichment is significant over a distribution of similar non-differentially expressed genes; however, we interpret this as CTCF binding having a small effect on gene expression directly. Kaushal et al 2021 also mention that they expected indirect transcriptional changes. 

We changed the sentence to the following: “The observed changes are slight however, which may explain why large-scale defects in transcription are not observed with RNA-sequencing in flies lacking dCTCF.”

6. Lines 80-81: “Differential expression in spatial regions by sequencing” was previously performed by the references cited in the introduction, and therefore stating that this “was previously only possible by mechanical manipulation” should be toned down.

The references cited in the introduction conducted single-cell RNA-sequencing after cellularization, therefore we added the words, “of embryos prior to cellularization” as well as changed “mechanical manipulation” to “slicing embryos” to be more specific. 

7. Line 169 must be amended (“nuclei ranked below the to the expected number of nuclei”).

8. Line 206: Figure references have typos.

9. Lines 218 and 220: First words of sentences should be capitalized.

We addressed points 7-9 by correcting the typos. 

10. I did not understand the meaning of the sentence in lines 234-236.

The original intention of this sentence was to illustrate that we could not confidently assign specific tissue fates (i.e. neuroectoderm, mesoderm, etc) to the different clusters so we assigned the clusters more generic identities (i.e. anterior, posterior, etc) instead. In order to avoid confusion here, we ultimately decided to delete this sentence. 

11. Cluster 9 should be discussed further, even if its spatial identity could not be determined. Why is this cluster only detected in WT but not CTCF[mat-] embryos?

We have speculated why cluster 9 is not present in CTCF[mat-] embryos; however we believe the true answer will require further investigation outside of the scope of this manuscript. We agree that it’s interesting to note at the very least, so we have added the following: 

“Interestingly, cluster 9 appears to be absent in dCTCFmat-/- embryos (Fig 1A-B). Without knowing the identity of cluster 9, we can only speculate why this may be the case; however, this raises the possibility that dCTCF may play a role in nuclear fate.” 

12. Fig. S4 panel C is not described in the figure legend.

We added “and (C) after removing low quality nuclei.” 

Reviewer #2: The authors Albright et al classified embryonic nuclei by single-nucleus RNA-seq and examined CTCF-regulated gene expression in these nuclei by comparing wild type to CTCF maternal null mutants during zygotic genome activation. They identified more cluster-specific differential expression than in bulk RNA-seq and highlighted several examples of differential expression of spatial marker genes in specific clusters. The work should be of general interest. However, the data requires further analyses, and the conclusions were not clearly presented or fully supported. Major revisions are needed to both the analyses and writing.

1. Figure numbering is incorrect for all supplementary figures, and it is not possible to understand which figures the authors were calling for. The intended Figure S1 is missing. There is no Figure S9 in the submission.

We have corrected this in the figure submission. 

2. The authors should indicate whether there is zygotic CTCF expression in this mutant. A diagram will be very helpful. The missing Figure S1 makes it more challenging to understand the properties of this KO.

We agree that the missing Figure S1 makes interpreting our mutant challenging. We have corrected that as well as further clarified the mutant in the text with the following: 

”We confirmed that our dCTCFmat-/- embryos lack maternal dCTCF at 0h and 2h after laying (S1 Fig). These experiments were conducted agnostic of the zygotic genotype given that zygotic genome activation largely does not occur until nuclear cycle 14 [1,2].”

3. Although the CTCF maternal knockout is known to be viable, dysregulation of Hox gene expression has been reported in embryos. The authors should characterize their new mutants and compare with previous data (Gambetta and Furlong, 2018) to report consistent or distinct phenotypes, e.g., viability and expression pattern of Hox genes.

Gambetta and Furlong 2018 do show that dCTCF is required for proper Abd-B expression; however, Kaushal et al 2021 (of which Dr. Gambetta is the principal author) report that their results in bulk RNA sequencing embryonic CNS do not show that Abd-B is differentially expressed upon the loss of dCTCF. This comment raises an excellent point in that dCTCF is potentially implicated in the regulation of Hox gene expression or topologically associating domains surrounding the Hox genes, therefore we have added a new figure (S10Fig) depicting expression in a select few Hox genes. 

We added the following text discussing this figure: “Because dCTCF is required for proper expression of Abd-B [26], a Drosophila Hox gene, we also examined expression of several Drosophila Hox genes within each cluster and in bulk. Upon the loss of maternal dCTCF, Antp and abd-A are differentially expressed in certain clusters, but not in bulk (S10 Fig). However, we found no evidence of differential expression of Abd-B, in agreement with bulk RNA sequencing in embryonic CNS dCTCF mutants [27].”

We limited the number of Hox genes in this figure out of space concerns; however, we found no evidence of differential expression in the other Hox genes (lab, pb, Ubx). 

4. Figure 1B indicates loss of cluster 9 in CTCF maternal KO. Is this because the cells are absent or because their gene expression changes and they are classified into other clusters? This can be determined by in situ of cluster 9-specific genes.

Reviewer #1 had a similar comment above, and in response we added the following sentences: 

“Interestingly, cluster 9 appears to be absent in dCTCFmat-/- embryos (Fig 1A-B). Without knowing the identity of cluster 9, we can only speculate why this may be the case; however, this raises the possibility that dCTCF may play a role in nuclear fate.” 

We acknowledge that this is interesting and would like to follow up with in situ hybridization among other experiments relating to CTCF function in early development; however, we believe that the optimization and execution of many in situ hybridizations is outside of the scope of this manuscript. We believe the addition of the statements above are sufficient to point out this peculiarity without over-speculating on the cause.

5. What is the accuracy of spatial prediction based on the RNA-seq? How many top marker genes were checked, and how many have consistent expression patterns with the in situ data?

This is a very important question that we appreciate you raising. We show additional markers in S7 Fig. We did check the top 20 marker genes as shown in S1 Table, as well as a few genes in the top 50 that we recognized by name. 

We do acknowledge that some of the marker genes in S1 Table do not exhibit patterned gene expression; however, we believe that this could be explained by a number of possibilities. For one example, small differences in expression of ubiquitously expressed genes can appear statistically significant enough for these genes to appear as markers. This particular example highlights the importance of double checking the expression patterns by in situ hybridization before calling spatial regions. As shown in the manuscript, we used existing in situ hybridizations from the Berkeley Drosophila Genome Project to confirm patterning. 

As mentioned previously, some of the marker genes in S1 Table do not show patterned gene expression; however, clusters 0-7 contained multiple spatially patterned genes beyond what was shown in Fig 2 and S7 Fig. Without conducting additional experiments, such as single-molecule FISH in control and dCTCF[mat-] embryos, we cannot give an exact estimate of accuracy or confirm patterning defects. However, the fact that several genes show patterning according to the spatial regions we called each cluster provided us with enough confidence to determine where the nuclei originated from within the embryo. 

6. The authors should verify their knockout by western blots, which are mentioned but not presented.

We corrected this with the addition of Figure S1. 

7. The authors stated that spatial identities cannot be assigned to clusters 8 and 9, but some quick searches with gene IDs in Figure 1C identified embryonic CNS for cluster 8 (maybe 9) and another clear distinct spatial pattern for cluster 9. The authors need to dig deeper into the spatial identity by searching more genes in those clusters using publicly available data.

This reviewer raises an excellent point about potential ambiguity in the way that we refer to spatial patterning, particularly in clusters 8 and 9. When double checking the Berkeley Drosophila Genome Project in situ hybridization database however, we noticed that many genes in clusters 8 and 9 show no staining in Stages 4-6. Stage 5 corresponds to nuclear cycle 14, the time point at which we collected our embryos. Here are a few examples of cluster 8 and 9 marker genes:

CG2233 - 

https://insitu.fruitfly.org/cgi-bin/ex/report.pl?ftype=3&ftext=GH20802-dg

CG2225 - 

https://insitu.fruitfly.org/cgi-bin/ex/report.pl?ftype=1&ftext=FBgn0032957

CG3408 - 

https://insitu.fruitfly.org/cgi-bin/ex/report.pl?ftype=1&ftext=FBgn0036008

The lack of staining in these images does not preclude the possibility that these genes are actually expressed in pre-cellularization nuclei at small levels or that these nuclei are fated to become the embryonic CNS; however, we cannot confidently identify these clusters without further investigation.

8. The authors need to indicate how much overlap there is between differential expression of different clusters. For example, cluster 9 has ~170 DE, cluster 3 has ~80 DE, but the common DE between clusters 3 and 9 is ~20 genes. Does this mean that most DE genes are cluster-specific? The authors need to quantify and present the data clearly. It is extremely hard to decipher the information presented in current Figure 3A.

9. The authors concluded that they identified more cluster-specific DE than in bulk but did not present data beyond the information in Figure 3A, horizontal bars. Do these bars include primarily the same genes, or they are totally different genes? It is very unclear how many more genes are detected as cluster-specific than in bulk. The authors state on page 12, “Many other genes are also differentially expressed in groups of clusters, but not in bulk.” This is very important and quantifiable information, and the authors need to present the data clearly.

We agree that we need to clarify the UpSet plot in Figure 3A, and hope that our edits (relevant changes italicized/underlined below for reference) to the figure legend satisfy the reviewer in resolving comments 8 and 9.

“(A) UpSet plot for visualizing the top 40 shared sets of candidate differentially expressed genes between control and dCTCFmat-/- nuclei within each cluster and in bulk. Horizontal bar plot (A, left) represents the total number of candidate differentially expressed genes within the cluster of the corresponding row. The vertical bar plot (A, top) represents the number of shared candidate differentially expressed genes for the conditions indicated below and is sorted from largest to smallest intersecting set, with each count representing a unique gene. Connected dots (black) represent the corresponding group of genes in the vertical bar plot above that might be differentially expressed in the clusters represented by rows with a filled in circle. Candidate genes differentially expressed in a single cluster are represented in blue.”

The horizontal bars do not count unique genes as they represent the number of differentially expressed genes in a single cluster, a single gene can be differentially expressed in any number of clusters. However, the vertical bar plot counts are of unique genes because each gene can only belong to one set (similar to how one would consider counts in a Venn diagram, UpSet plots being a more recent solution to data with many intersecting sets). 

10. The authors need to verify their RNA-seq data with another type of assay, e.g., RNA FISH. The differential expression (e.g., Figure 3B and C) should be verified between wildtype vs. CTCF KO cells, and between wildtype clusters of cells. Without such verification, it is hard to conclude that the single nucleus RNA-seq provides useful information. The verification should be done at least for the top differences.

We completely agree that RNA FISH will be necessary to verify differential expression; however, we have edited our intention for our manuscript in the abstract, “In order to establish the use of single-nucleus RNA sequencing in Drosophila embryos prior to cellularization, here we look at gene expression in control and insulator protein, dCTCF, maternal null embryos during zygotic genome activation at nuclear cycle 14.” 

While we acknowledge that CTCF is an interesting case of biology, from the beginning this manuscript was intended to establish the use of single-nucleus RNA sequencing in pre-cellularization embryos. Rather than address the many questions surrounding the role of CTCF in early development, we hope that our work serves as a resource for people to follow-up on.

At the end of our manuscript, we also state: “Whether or not the changes in gene expression that we observed have implications in embryonic development related to the loss of dCTCF is unclear without further investigation, such as single-molecule FISH to validate the observed changes in gene expression of particular RNAs. Ultimately, using single-nucleus RNA-sequencing to examine changes in gene expression upon the loss of important developmental factors has the potential to uncover perturbation responses previously undetected by bulk RNA-sequencing.” 

We believe that our statements in the introduction and discussion, with additional changes to language throughout the manuscript deemphasizing true differential expression, are sufficient enough to address the reviewer’s concerns that we do not have FISH data in this manuscript as we highlight this work more for its potential to aid future studies. 

11. How do the authors explain that most DE is up-regulation? Does this agree with Kaushal et al., 2021? It seems likely that decreases in DE are still masked by the inability to further separate cell types using this approach.

We believe that this is an interesting point to note; however, we can only speculate as to why most differentially expressed genes appear to be up-regulated and decided not to comment on this in the manuscript. Kaushal et al. 2021 did report almost twice as many up-regulated genes compared to down-regulated genes in larval CNS (Figure 3a in their manuscript). 

We would argue that spatial differential expression is unmasked in our ability to cluster nuclei into distinct groups despite their belonging to a syncytium where no cells are present. Because we have prior knowledge of spatial expression in embryonic nuclei, we were able to examine the spatial identities of clusters over many iterations of our pipeline and are confident in our data. If we were to over-cluster, or further separate our nuclei into more clusters, we run the risk of highlighting non-biological variation as something biological. 

Global effects on gene expression are apparent in our pseudo-bulking results from individual nuclei, as well as in Kaushal et al. 2021 in bulk extractions and sequencing of larval CNS RNA. We believe that our results in Figure 3 are sufficient to show that we have unmasked local changes in differential expression. We acknowledge that the examples we chose for Figure 3B-D are all up-regulated, but maintain that this is valid regardless of whether genes are up or down-regulated. 

12. Lines 269-272: “Because we found many differentially expressed genes, we considered that this may be due to low expression given the sparsity of single-nucleus RNA-sequencing; however, we found that the mean expression of differentially expressed genes in single or multiple clusters overall does not have a substantially different pattern from that of non-differentially expressed genes.” The authors should compare and present mean expression of differential genes versus non-differential genes. This analysis is essential to rule out the possibility that more differential genes in cluster data than bulk data result from inaccurate quantification due to insufficient sequencing and coverage. Figure S9 may contain such information but is currently missing. The authors should also clearly define what they mean by “a substantially different pattern”.

The addition of missing S1 Fig corrected the misnumbering of our figure. S9 Fig shows a comparison of mean expression in non-differentially expressed genes (not DE), genes differentially expressed in multiple clusters (multi DE), and genes differentially expressed in a single cluster (single DE). 

To clarify what we mean by ‘substantially different pattern,’ we added: “Each of these curves are right-skewed, or most genes are expressed in low levels at less than 100 transcripts per million (TPM).” 

13. Figure 3B-D: the expression change of patterning markers may lead to morphogenesis defects – did the authors examine the tissue morphology and distribution of marker gene expression by in situ or RNA FISH in embryos/larvae? Is the difference caused by strong differences in a small group of cells or weak differences in all cells in one cluster?

14. What are the most affected factors and signaling pathways in CTCF KO? How many of these genes are CTCF binding targets based on published CTCF ChIP-seq?

We thank you for comments 13 and 14 which allude to future studies on CTCF function in early embryonic development. Kaushal et al. 2021 does contain examples of local changes in expression upon loss of CTCF, but later than our study during embryonic development. Gambetta and Furlong 2018 show a slight change in the domain of Abd-B expression in addition to morphological changes, also later in development. 

When hand-sorting embryos prior to isolating nuclei for these experiments, we did not observe any morphological defects in the pre-cellularization embryos. While optimizing our experiments we did keep old plates containing CTCF[mat-] embryos, and concur that maternal CTCF is not required for embryogenesis. Beyond this observation, we did not conduct careful morphological analyses of our CTCF[mat-] mutant. 

Our main objective for this study is to highlight the potential for single-nucleus RNA-sequencing to complement studies in patterning prior to cellularization. We agree that understanding the biology of CTCF, as well as other insulators, is important and inherently interesting; however, outside of the scope of this manuscript. We decided to use a dCTCF mutant in this manuscript due to general interest in the protein, but do not intend for this manuscript to answer all of our questions on its function in early Drosophila embryonic development. We hope that our data will be useful to those conducting more careful studies into CTCF function and effects on patterning in the early embryo. 

15. Please finish the sentence in lines 141, 169 (“to” and “the expected number of *UMI*”?).

We have corrected these mistakes in the updated manuscript.

---

## [Decision Letter · Decision Letter 1]

30 May 2022

PONE-D-22-01112R1Single-nucleus RNA-sequencing in pre-cellularization Drosophila melanogaster embryosPLOS ONE

Dear Dr. Albright,

Thank you for submitting your revised manuscript to PLOS ONE. Your revision has now been reevaluated by one of the original reviewers. As you will see the reviewer feels that your revision has further improved your study. There are a few points that would need to be addressed before publication can be considered. This likely would need no further experiments but changes to the conclusions and positioning of your study. If you would be able to address the remaining points and send a further revised version of your manuscript along with a point-to-point response to the reviewer's points I would be in the position to make a decision on publication.

We look forward to receiving your revised manuscript.

Kind regards,

Anton Wutz

Academic Editor

PLOS ONE

Journal Requirements:

Reviewers' comments:

Reviewer's Responses to Questions

**Comments to the Author**

1. If the authors have adequately addressed your comments raised in a previous round of review and you feel that this manuscript is now acceptable for publication, you may indicate that here to bypass the “Comments to the Author” section, enter your conflict of interest statement in the “Confidential to Editor” section, and submit your "Accept" recommendation.

Reviewer #1: (No Response)

2. Is the manuscript technically sound, and do the data support the conclusions?

Reviewer #1: Yes

3. Has the statistical analysis been performed appropriately and rigorously? 

Reviewer #1: Yes

4. Have the authors made all data underlying the findings in their manuscript fully available?

Reviewer #1: Yes

5. Is the manuscript presented in an intelligible fashion and written in standard English?

Reviewer #1: Yes

6. Review Comments to the Author

Reviewer #1: Most of my comments were addressed by the revisions, except for a few points below to be clarified.

1. Lines 212-213: Related to my original points 1+2, this sentence could cause confusion if the reader interprets this as meaning that snRNA-seq was performed prior to zygotic genome activation. In this study, snRNA-seq was performed on embryos undergoing zygotic genome activation; therefore, dCTCF[mat-/-] embryos probably transcribe CTCF mRNA (this could be verified in the snRNA-seq data) but do not translate (at least high levels of) CTCF protein yet. This could be explained more clearly.

2. Lines 56-60: Related to my original point 6, I feel that the authors should tone down their argument that previous scRNA-seq studies may not have primarily measured zygotic expression due to the presence of maternal cytoplasmic RNAs in cells. The present study, Karaiskos et al. 2017, and Ing-Simmons et al. 2021 likely all handled nuclei instead of cells because embryos were dounce-homogenized in all three studies – resulting in nuclei, not intact cells. In my view, the information obtained by snRNA-seq and scRNA-seq in fly embryos is thus comparable (if the authors don’t agree, please explain why). The novelty of the current study is, in my view, rather the demonstration that snRNA-seq is sensitive enough to detect relatively subtle gene misexpression defects in mutant embryos lacking major defects in cell fate decisions.

3. Line 317: “Embryonic” should be changed to “larval”, as RNA-seq was performed on central nervous systems of third instar larvae (not embryos) in Kaushal et al. 2021.

7. PLOS authors have the option to publish the peer review history of their article (what does this mean?). If published, this will include your full peer review and any attached files.

Reviewer #1: No

---

## [Author Response · Author response to Decision Letter 1]

7 Jun 2022

Reviewers' comments:

Reviewer's Responses to Questions

Comments to the Author

1. If the authors have adequately addressed your comments raised in a previous round of review and you feel that this manuscript is now acceptable for publication, you may indicate that here to bypass the “Comments to the Author” section, enter your conflict of interest statement in the “Confidential to Editor” section, and submit your "Accept" recommendation.

Reviewer #1: (No Response)

2. Is the manuscript technically sound, and do the data support the conclusions?

Reviewer #1: Yes

3. Has the statistical analysis been performed appropriately and rigorously?

Reviewer #1: Yes

4. Have the authors made all data underlying the findings in their manuscript fully available?

Reviewer #1: Yes

5. Is the manuscript presented in an intelligible fashion and written in standard English?

Reviewer #1: Yes

6. Review Comments to the Author

1. Lines 212-213: Related to my original points 1+2, this sentence could cause confusion if the reader interprets this as meaning that snRNA-seq was performed prior to zygotic genome activation. In this study, snRNA-seq was performed on embryos undergoing zygotic genome activation; therefore, dCTCF[mat-/-] embryos probably transcribe CTCF mRNA (this could be verified in the snRNA-seq data) but do not translate (at least high levels of) CTCF protein yet. This could be explained more clearly.

We agree that we could better explain the timing of our collection and the assumptions we were working under. We added language to the figure legend for S1 Fig, “The 2h embryos were aged for an additional 2 hours with the majority of the embryos representing nuclear cycle 14, the same time point at which we conducted single-nucleus RNA-sequencing,” in order to clarify that we did not detect CTCF protein at this time point. 

As you have alluded to, we are working under the assumption that any zygotically expressed CTCF would not have time to be translated, folded, and function. For this reason, we never intended to genotype the embryos for zygotic dCTCF. In the figures pasted below (see attached document), we do see low CTCF expression (average 0.81 TPM); however, we only detected expression in 270 control and 467 nuclei (less than 10% of our total nuclei). Although expression appears higher in the dCTCF[mat-/-] embryos and in individual clusters by eye, these differences are not statistically significant. We decided not to include these panels in S1 Fig as the Western blot definitively shows the absence of dCTCF protein in the early embryo. 

As far as what we originally had in 212-213, we decided to remove those sentences entirely in order to avoid the confusion. We believe that the Western blot is enough to show that dCTCF protein will have no effect on our results, because we did not detect it, without speculating on the effect of zygotic gene expression. 

2. Lines 56-60: Related to my original point 6, I feel that the authors should tone down their argument that previous scRNA-seq studies may not have primarily measured zygotic expression due to the presence of maternal cytoplasmic RNAs in cells. The present study, Karaiskos et al. 2017, and Ing-Simmons et al. 2021 likely all handled nuclei instead of cells because embryos were dounce-homogenized in all three studies – resulting in nuclei, not intact cells. In my view, the information obtained by snRNA-seq and scRNA-seq in fly embryos is thus comparable (if the authors don’t agree, please explain why). The novelty of the current study is, in my view, rather the demonstration that snRNA-seq is sensitive enough to detect relatively subtle gene misexpression defects in mutant embryos lacking major defects in cell fate decisions.

This is a great point, we were relying on the assumption that the other works isolated cells rather than nuclei, but realize that this may not necessarily be the case. In order to tone down the language in our manuscript, we removed the statements about zygotic gene expression in lines 56-60 and replaced them with the underlined, “These studies demonstrate the potential for single-cell RNA-sequencing to answer questions relating to pattern and body axis formation in the early Drosophila embryo; however, whether single-cell RNA-sequencing is sensitive enough to detect subtle changes in gene expression in mutant embryos lacking major defects remains unclear.” 

We believe this change still highlights that our work was conducted earlier in development, as well as detects subtle changes in expression as you mentioned, but removes the unclear language on zygotic gene expression. 

3. Line 317: “Embryonic” should be changed to “larval”, as RNA-seq was performed on central nervous systems of third instar larvae (not embryos) in Kaushal et al. 2021.

Thank you for catching this, we have fixed this in the manuscript.

---

## [Editor Report · Decision Letter 2]

13 Jun 2022

Single-nucleus RNA-sequencing in pre-cellularization Drosophila melanogaster embryos

PONE-D-22-01112R2

Dear Dr. Albright,

thank you for sending your further revised manuscript. I have now read through your revision and answers to the remaining points raised by reviewer 1. Your revision has succeeded to address all remaining concerns in a satisfactory manner and your study is now scientifically suitable for publication. Your manuscript will be formally accepted for publication once it meets all outstanding technical requirements.

Kind regards,

Anton Wutz

Academic Editor

PLOS ONE
---

## [Editor Report · Acceptance letter]

17 Jun 2022

PONE-D-22-01112R2 

Single-nucleus RNA-sequencing in pre-cellularization *Drosophila melanogaster* embryos 

Dear Dr. Albright:

I'm pleased to inform you that your manuscript has been deemed suitable for publication in PLOS ONE. Congratulations! Your manuscript is now with our production department. 

Kind regards, 

on behalf of

Dr. Anton Wutz 

Academic Editor

PLOS ONE